# Anti-VEGF Drugs in the Treatment of Multiple Myeloma Patients

**DOI:** 10.3390/jcm9061765

**Published:** 2020-06-06

**Authors:** Roberto Ria, Assunta Melaccio, Vito Racanelli, Angelo Vacca

**Affiliations:** Department of Biomedical Sciences and Human Oncology, Section of Internal Medicine and Clinical Oncology, University of Bari “Aldo Moro” Medical School, 70124 Bari, Italy; assuntamel@hotmail.it (A.M.); vito.racanelli1@uniba.it (V.R.); angelo.vacca@uniba.it (A.V.)

**Keywords:** angiogenesis, microenvironment, multiple myeloma, vascular endothelial growth factor, vascular endothelial growth factor receptor

## Abstract

The interaction between the bone marrow microenvironment and plasma cells plays an essential role in multiple myeloma progression and drug resistance. The vascular endothelial growth factor (VEGF)/VEGF receptor (VEGFR) pathway in vascular endothelial cells activates and promotes angiogenesis. Moreover, VEGF activates and promotes vasculogenesis and vasculogenic mimicry when it interacts with VEGF receptors expressed in precursor cells and inflammatory cells, respectively. In myeloma bone marrow, VEGF and VEGF receptor expression are upregulated and hyperactive in the stromal and tumor cells. It has been demonstrated that several antiangiogenic agents can effectively target VEGF-related pathways in the preclinical phase. However, they are not successful in treating multiple myeloma, probably due to the vicarious action of other cytokines and signaling pathways. Thus, the simultaneous blocking of multiple cytokine pathways, including the VEGF/VEGFR pathway, may represent a valid strategy to treat multiple myeloma. This review aims to summarize recent advances in understanding the role of the VEGF/VEGFR pathway in multiple myeloma, and mainly focuses on the transcription pathway and on strategies that target this pathway.

## 1. Introduction

Multiple myeloma (MM), a hematological cancer, accounts for about 1% of all human tumors, and is characterized by the infiltration of rich bone marrow (BM) by mature plasma cells (PCs) that produce monoclonal immunoglobulins [1,2,3]. The clonal PCs produce and release cytokines that are responsible for the typical clinical manifestation of the disease: (i) bone resorption (lytic lesions, hypercalcemia, bone pain) caused by alteration in the activity of osteoclasts/osteoblasts; (ii) anemia caused by modification of the maturation and differentiation of erythroblasts; (iii) renal insufficiency due to Ig light chain deposition; (iv) hypercalcemia and hyperuricemia; and (v) hyper-viscosity syndrome caused by high circulating protein levels [4].

MM progression is accompanied by and strictly dependent on changes in the microenvironment of the BM [5,6]. These modifications of the microenvironment induce a permissive environment that protects and stimulates plasma cell survival and proliferation [5,6,7,8]. The interaction of MM PCs with BM stromal cells (SCs) and extracellular matrix (ECM) components in the BM microenvironment is mediated by a plethora of autocrine and paracrine cytokine loops, as well as direct cell–cell and cell–ECM interactions. These direct and indirect interactions result in the activation of multiple signaling pathways that are responsible for modifications in the microenvironment during MM progression [9] and that are responsible for MM plasma cell apoptosis inhibition, survival, proliferation, and invasion. The expansion of neoplastic PCs in BM causes bone lysis and promotes microenvironment modulation and neovessel formation [10,11,12].

MM-associated microenvironment modifications include BM neovessel formation for assembling the “vascular niche” and bone cell activation for the constitution of the “osteoblastic niche” [13]. In these two specialized niches, myelomatous PCs grow, survive, and are protected from external attacks [13,14]. The alterations occurring in these niches represent predisposing events that facilitate the survival and expansion of neoplastic PCs. Moreover, the cells that comprise these specialized niches contribute to protecting MM PCs from the aggression of chemotherapy and immunological cells. The elucidation of key niche-associated pathways, including the primary driver of mutations in BM stromal cells, the role of hypoxia, angiogenesis, and inflammation can increase our knowledge of immune evasion and activation of survival pathways, and could indicate ways to improve modern therapeutic approaches [14].

The development of a new vascular tree in the BM of MM patients is a pathologic process in which angiogenesis (the formation of new vessels from existing ones) [9], vasculogenesis (the formation of new vessels from endothelial precursors) [15], and vasculogenic mimicry (the completion of neovessels by other non-endothelial cells [ECs]) [16] work simultaneously for the constitution of the vascular niche [9]. BM neovascularization is related to the MM stage, disease progression, and patients’ response to therapy and survival [15,16,17,18,19,20].

Taken together, these processes lead to modifications in the BM microenvironment and its controllers (activated cells, cytokines, and their autocrine and paracrine loops, signaling pathways), which are useful targets in the treatment of MM, for example, the direct targeting of MM PCs [5,21,22].

## 2. Angiogenesis in MM Progression

### 2.1. The Bone Marrow Microenvironment

The components of the BM microenvironment (SCs and ECM) surround MM PCs and support them by direct cell–cell and cell–ECM interaction, and by the production of cytokines and growth factors [23,24].

BM ECs express adhesion molecules and receptors on their cell surface, which is characteristic of a typical “activated” phenotype [25]. This activated phenotype is related to a specific genotype of MM BM ECs [26], unlike those of monoclonal gammopathy of undetermined significance (MGUS) or normal resting ECs. Phenotypic and genotypic activation causes the quick and uncontrolled proliferation of ECs, and angiogenesis self-maintenance [27,28,29,30,31]. Activated MM ECs modulate the expression of receptors, increasing VEGF receptor (VEGFR)-2, tyrosine-protein kinase Met (cMet, also called hepatocyte growth factor receptor), fibroblast growth factor receptor (FGFR), and Tie2/Tek density, integrins, and other adhesion molecules responsible for adhesion to the ECM components and cell motility. Moreover, integrin-activated signaling, particularly the β3-integrin pathway, sustains cell survival, proliferation, migration, and capillarogenesis. The interaction of MM PCs and activated ECs is mediated by endoglin and favors PCs entrance into neovessels. Finally, MM ECs express aquaporin 1 on their membrane, which is a water transporter that is responsible for plasma extravasation and increases interstitial pressure by increasing vascular permeability [25,32,33]. Aquaporin 1 also upregulates hypoxia-inducible Factor-1 alpha (HIF-1α) and vascular endothelial growth factor (VEGF) because of hypoxia induction [25,32,33].

The recruitment of progenitor cells to differentiate into a subset of mature ECs is demonstrated by the expression on their surface of the CD133 staminal antigen, which shows active vasculogenesis [7,29,30,31,32,33,34,35,36,37,38,39]. The down or upregulation of some genes like tumor cells and cells with a well-defined proteomic signature render MM ECs different from MGUS and other resting ECs, characterizing their over-angiogenic, and transformed phenotype [34].

The presence of fibroblasts, also called “activated myofibroblasts” or “cancer-associated fibroblasts”(CAFs) that are derived from cells undergoing the endothelial-mesenchymal or mesenchymal transition has been demonstrated in the BM of patients with active MM [6,40,41,42,43]. The precursors of CAFs are resident fibroblasts and progenitor cells. BM infiltrating fibroblasts promote neovessel formation by the secretion of high amounts of growth factors and angiogenic cytokines, such as VEGF-A, FGF-2, tumor necrosis factor-alpha (TNF-α), urokinase, and matrix metalloproteinases–MMPs [7,9,13,40,41,42,43]. Moreover, fibroblasts synthesize inducible nitric oxide synthase, which increases blood flow [41].

Immune changes during MM progression play an essential role in the BM microenvironment. The distribution and phenotypic and behavioral features of the natural killer cells, cytotoxic and other T cells, monocytes, and macrophages play an essential role in the progression from premalignant MM to active disease. A precise understanding of these modifications will help to develop new therapeutic strategies as well as accurate patient stratification [44].

Circulating monocytes are recruited to differentiate into active macrophages in the BM of MM patients through the activation of VEGF-A/VEGFR-1 and FGF-2/FGFR-1, -2, and -3 pathways. Here they acquire phenotypic and functional adaptations that contribute to the completion of neovessel walls through vasculogenic mimicry [16]. They also secrete VEGF and FGF-2, which play a role in promoting and auto-maintaining angiogenesis, vasculogenesis, and vasculogenic mimicry. Mast cells are also recruited in the BM by stem cell factor (SCF), FGF-2, VEGF-A, and platelet-derived growth factor (PDGF) secreted by MM PCs and other SCs [18,38,39]. Mast cells release angiogenic factors (VEGF-A, FGF-2, transforming growth factor (TGF)-β, TNF-α, interleukin (IL)-8, heparin, histamine, tryptase, and chymase) by degranulation [38,45]. Moreover, like macrophages, mast cells contribute to vasculogenesis mimicry by phenotypic adaptation [38,45,46]. They maintain typical tryptase-positivity and acquire endothelial cell-like properties. In the BM of MM patients, mast cells interact physically with ECs participating in the completion of the vascular wall [38,45,46].

MM PCs unbalance the osteoblasts/osteoclasts equilibrium and this induces bone disease [13,47,48,49]. In the BM niches, there is an interplay between MM PCs, osteoclasts, and vascular ECs [13]. This close link induces a vicious circle between angiogenesis, bone lysis, and favors BM myeloma cells survival and expansion [13,48,49].

Hematopoietic stem cells (HSCs–CD133+) and endothelial progenitor cells (EPCs-CD146+ CD105+ CD34+) residing in the BM and recruited from other sites, are induced to differentiate into mature ECs by angiogenic cytokines (VEGF-A, FGF-2, insulin-like growth factor (IGF-1)) released in the BM. This biological process, known as vasculogenesis, is an ancestral process that is active during prenatal life, and has been demonstrated to be reactivated in the BM of MM patients with active disease [50,51,52,53]. 

Mesenchymal stem cells (MSCs) are one of the main components of BM niches [15,54,55,56,57]. Their origin remains unclear, but MSCs can differentiate into fibroblasts, adipocytes, chondrocytes, and osteoblasts [54,56,57,58]. MSCs support MM plasma cell growth, survival, and immune system evasion [54,55,56]. 

Lastly, high amounts of tumor-infiltrating adipocytes have been found in the BM of MM patients [59,60]. BM adipocytes sustain angiogenesis by secretion of VEGF, Angiopoietin (Ang)-1 and -2, leptin, adiponectin, TNF-α, FGF, TGF-β, hepatocyte growth factor (HGF), IL-6, and IL-8 [59,60]. The hypoxic environment of MM BM further stimulates tumor-infiltrating adipocytes to produce angiogenic factors, particularly VEGF-A, which have been demonstrated to promote disease progression through paracrine and autocrine loops in the BM of MM patients [57,58,59,60].

### 2.2. Angiogenesis in MM

BM neovessels in MM patients have an impaired structure [15]. The vascular tree is formed by highly permeable and fenestrated thin, tortuous, and arborized blood vessels. The ECs that line newly formed vessels contain cytoplasmic vesicles and inclusions, broaden intercellular junctions, and are supported by an incomplete and highly permeable basement membrane [15]. These alterations are a result of microenvironment activation and stimulation induced by myeloma PCs, which promotes three different processes: (i) angiogenesis, (ii) vasculogenesis, and (iii) vasculogenic mimicry [9].

BM microvessel density is remarkably higher in patients with active MM compared to those with MGUS or healthy subjects and neovessel density correlates with the disease stage, shrinking during the remission/plateau phase, increasing in density in the diagnosis, relapse/refractory phase, and reaching maximum expansion in the leukemic phase [9,11,32,58]. 

Angiogenesis, that is, the rise of neovessels from already formed and stable ones, is the primary, active biological process in the BM of MM patients [9,11,32,58]. Angiogenesis is essential for cancer cell proliferation, spread, and diffusion. Angiogenesis starts when the preneoplastic disease (MGUS), which is characterized by plasma cell growth equilibrium and poor vascularization (avascular phase), and progresses to active proliferation and spread of PCs with uncontrolled and unlimited neovessel formation (the vascular phase). MM progression is related to the activation of many oncogenes (i.e., c-myc, c-fos, c-jun, ets-1, etc.), which is induced by immunoglobulin translocations and genetic instability [12,32,57]. The activation of angiogenesis in this phase is due to the release of angiogenic cytokines, such as VEGF-A, FGF-2, HGF, etc., by transformed PCs [12,32,58].

Like angiogenesis, vasculogenesis contributes to neovascularization in the BM of MM patients [28,29,30]. This biological process is active during prenatal life, where it induces the primary development of the yolk sac vasculature, the heart, and the dorsal aortae during embryogenesis [27]. The aggregation of EPCs (angioblasts) in a primitive capillary plexus stored in the mesoderm and their differentiation into mature ECs are the basis for the formation of the new vascular system [27]. During neovascularization in the BM of MM patients, this ancestral biological process is reactivated to release angiogenic cytokines into the microenvironment. It has been demonstrated that peripheral blood-purified CD34+ VEGFR-2+ cells from MM patients exposed to VEGF-A, FGF-2, and IGF, in appropriate culture conditions, acquire a typical morphology (spindle-shaped) and change their phenotype to acquire typical EC markers such as factor VIII-related antigen (FVIII-RA), vascular endothelial-cadherin (VE-cadherin), CD34, CD31, VEGFR-2, Tie-2/Tek, and E-selectin [29]. These in vitro results have been confirmed in the BM of MM patients where some neovessel forming ECs express the CD133 staminal antigen on their surface, like the typical EC markers [29]. The expression of the CD133 staminal antigen was absent in MGUS patients. Moreover, several studies have demonstrated the presence of circulating EPCs in MM patients [28,29]. 

Vasculogenesis mimicry is another pathological process of neovascularization that is found in aggressive solid and hematologic tumors. In this particular process, malignant cells and/or other non-ECs, such as inflammatory cells, acquire the capacity to assemble a vessel-like network [16,30,38,39,59]. The Vasculogenic mimicry is a process in which epithelial-mesenchymal transition plays a very important role. Cancer cells, cancer stem cells, inflammatory cells, tumor-associated fibroblasts, under the stimulus of angiogenic cytokines (FGF-2, VEGF-A, HGF, IGF), inflammatory cytokines (IL-6 and -8, TGF-α and -β, TNF-α, CCL-2), and other signaling pathways (stromal cell-derived factor 1 (SDF1), cyclooxygenase-2 (COX-2), Twist, E-cadherin, EPH receptor A2 (EphA2)-phosphatidylinositol 3-kinase (PI3K), Wnt, and Notch signaling pathway) undergo an epithelial–mesenchymal transition by downregulation of E-cadherin, zonula occludins-1, and α-catenin, and upregulation of VE-cadherin, fibronectin, cadherin-2, and vimentin. HIFs and hypoxia-responsive elements (HREs) play an important role in this context. In MM, inflammatory cells (i.e., macrophages and mast cells) isolated from the BM of patients with active disease contribute to BM neovascularization by vasculogenic mimicry. In appropriate culture conditions, when these cells are exposed to angiogenic cytokines (VEGF and FGF-2), they modify their phenotype and acquire biologic properties similar to those of ECs producing functional capillary-like structures in vitro [16,30,38,39,59]. Vasculogenic mimicry mediated by inflammatory cells has been demonstrated in the BM biopsies of MM patients [16,38].

## 3. The VEGF/VEGFR Pathway

VEGF is a member of a family of six structurally related proteins, namely, VEGF-A, -B, -C, -D, -E (viral factor), and PDGF [61,62]. Their activity is mediated by interaction with the relative receptor and mediate angiogenesis (VEGF-A, -E/VEGFR-2-neuropilin (NRP)-1, -2), or lymphangiogenesis (VEGF-C, -D/VEGFR-2, -3) [58,59]. The role of the bind VEGF-B/VEGFR-1 is unclear (Figure 1).

Ferrara and Henzel first purified VEGF, a glycoprotein with mitogenic activity for vascular ECs, from bovine pituitary folliculo-stellate cells [62,63]. The VEGF family is composed of six members: VEGF-A, -B, -C, -D, -E, and placental growth factor (PlGF) that recognize and specifically bind three different transmembrane receptor tyrosine kinases, which are VEGFR-1, -2 and -3 (Figure 1). The interaction between VEGFs and VERFRs promotes EC activation and endothelium regeneration (Figure 2). Moreover, VEGF-A, -B, and PlGF/VEGFR-1 interaction increase vascular permeability (Figure 2). The activation of VEGFR-2 (KDR or Flk-1 is predominantly expressed in vascular ECs) by interaction with its ligands (VEGF-A, -C, -D) mediates VEGF-dependent angiogenesis and increases vessel permeability [63,64]. VEGFR-3 is restricted to lymphatic ECs, and the binding of VEGF-C and VEGF-D mediates the regulation of lymphangiogenesis (Figure 2) [64,65].

The tyrosine kinases’ VEGFRs are differentially expressed in various cell types [66,67,68,69]. The main member of the family, VEGFR-2, is prevalently expressed in ECs, neuronal, megakaryocytes, and hematopoietic stem cells [68], VEGFR-1 is expressed on the hematopoietic cell surface, monocytes/macrophages and smooth muscle cells [66,67]. The heterodimerization of the VEGFR-1 and -2 subunit mediates the binding of VEGF-B and PlGF (Figure 2) [67]. The activation of VEGFR-2 by VEGF-A on ECs activates a tyrosine kinase pathway, via protein kinase C (PKC) or the Ras protein, that terminally activates the mitogen-activated protein kinase (MAPK) system [69]. 

Differently from other cytokines (i.e., fibroblast growth factor–FGF-2, and PDGF), VEGF-A shows specific activity in vascular ECs, which are a specific growth factor for angiogenesis. Other cytokines act on different cytotypes and have several effects on biological activity that is different from angiogenesis. Other activities modulated by VEGF-A include the stimulation/differentiation of hemopoietic stem cells, ECM remodeling, and modulation of inflammatory cytokines [30,69,71]. VEGF-A secretion has been observed in several leukemic cell lines as well as in primary cancer cells, including MM cells [25,72,73]. Moreover, in these cells, paracrine and autocrine loops have been reported and shown to modulate the malignant behavior of hematological cancer cells [70,71,72,73]. In MM cells, as in other leukemic cells, VEGF-A activates its specific receptor expressed by EC, inducing the production of growth factors that stimulate leukemic cells to proliferate and induce drug resistance [74,75,76,77]. Moreover, the constitutive activation of VEGFR-2 on ECs of BM-derived from MM patients has been described previously [25,26,72,73].

Some metabolic factors, such as hypoxia and hypoglycemia, regulate the production and release of VEGF-A [78]. HIF-1 is a crucial angiogenesis triggering factor because it induces the expression of VEGF mRNA [79,80]. The nuclear-stabilized HIF-1α forms an active complex with p300 and phosphorylated-STAT3. This active complex induces the activation of the VEGF promoter upregulating RNA polymerase II-dependent VEGF transcription [78,79,80,81]. 

An increase in VEGF-A production and secretion by lysophosphatidic acid (LPA) through the activation of c-Jun N-terminal kinase (JNK) and nuclear factor kappa-light-chain-enhancer of activated B cells (NF-κB) has been reported [82]. Several studies indicate that angiogenesis is strictly dependent on Akt/NF-κB activation [83,84]. In hematopoietic cancer cells, the inhibitors of the NF-κB pathway induce VEGF secretion inhibition, thus reducing EC proangiogenic activities [85]. NF-κB is also activated by the PI3K/Akt signaling pathway, suggesting that PI3K/Akt activation plays a role in angiogenesis and leukemia progression [86].

VEGF expression is also modulated by several transcription factors, such as activator protein-1 (AP-1), NF-κB, and stimulatory protein-1 (SP-1) [87,88,89]. Among those, AP-1 is a critical factor in the modulation of VEGF-A gene transcription in several hematopoietic and solid cancer cells [90,91,92].

Published data indicate that VEGF-A directly stimulates the proliferation of cancer cells. In MM, VEGF mainly stimulates plasma cell migration, proliferation, and survival via autocrine and paracrine VEGF-A/VEGFR-2 loops (Figure 2) [9,93]. Also, leukemic cell resistance to apoptosis induced by serum deprivation is caused by VEGF-A by means of the expression of heat shock protein 90 (Hsp90), which binds B-cell lymphoma 2 (Bcl-2) and apoptotic protease activating factor-1 (Apaf-1) [94]. Moreover, EC exposure to VEGF-A induces the production and release of several hematopoietic growth factors including granulocyte-colony stimulating factor (G-CSF), granulocyte-macrophage colony-stimulating factor (GM-CSF), stem cell factor (SCF) and IL-6, which act as growth factors for hematopoietic cancers including MM [94,95]. It has been reported that VEGF-A expression is related to disease stage and tumor burden [96,97,98].

The migration of MM PCs during disease progression as well as of ECs during angiogenesis is related to ECM lytic enzymes, including matrix metalloproteinase (MMP)-2 and -9, and urokinase-type plasminogen activator (uPA) [98]. The production of ECM lytic enzymes by PCs and BM SCs is modulated by VEGF-A/VEGFR-2 engagement and activation of relative tyrosine kinase pathway [75,98,99].

VEGF-C is implicated in tumor lymphangiogenesis by activation of its specific receptor VEGF-R3 expressed in lymphatic ECs (Figure 2) [65]. VEGF-C acts by c-Jun binding and activation of the promoter of the cyclic adenosine 3′,5′-monophosphate-response element of the cyclooxygenase-2 (COX-2) inducing COX-2 expression [100,101,102]. COX-2 enhances cancer cell survival and proliferation and inhibits anti-tumor immunity. COX-2 upregulation is also due to VEGF-R2 activation, which induces p38 MAPK/JNK signaling pathways in human vascular ECs [101]. On the other hand, VEGF-A may also influence the upregulation of COX-2 in cancer cells. So, cancer tumor angiogenesis may be increased through the induction of COX-2/prostanoids produced by VEGFs (Figure 2) [102].

## 4. VEGF/VEGFR Inhibition in MM

Improved outcomes have been obtained for MM patients thanks to the addition of “target” biological drugs active in the microenvironment to the therapeutic strategies of MM [9,103,104,105].

Antiangiogenic activity, as well as the inhibition of BM SCs, has been demonstrated for proteasome inhibitors, immunomodulators (IMIDs), bisphosphonates, monoclonal antibodies, tyrosine kinase inhibitors, PI3K/Akt-MEK/ERK pathway inhibitors, focal adhesion kinase (FAK) inhibitors, interleukin inhibitors, farnesyltransferase inhibitors, and marine cartilage extract [106,107,108,109,110].

### 4.1. Preclinical Evidence on the Anti-Angiogenic Effect of Currently Used Therapy

#### 4.1.1. Proteasome Inhibitors

Several preclinical studies have described the antiangiogenetic activity of proteasome inhibitors (PIs). The first-generation PI, bortezomib inhibits the proliferation and migration of human umbilical venous endothelial cells (HUVEC), EC lines, and primary isolated BM ECs from MM patients. It downregulates several angiogenic cytokines, such as VEGFs, FGF-2, IGF-1, interleukins, and the expression of angiopoietins by MM PCs, ECs, and other SCs (fibroblasts and inflammatory cells) [111,112]. Moreover, bortezomib induces EC apoptosis, reduces the proliferation and motility of MM ECs, and inhibits neovessel formation in vitro [113,114]. The antiangiogenic activities of bortezomib are mediated by the inhibition of IKB degradation and the consequent block of NF-κB activity (Figure 3) [115,116]. Moreover, the block of NFκB causes inhibition of the expression and release of VEGF-A mediated by the stabilization of HIF-1α [117] and the CXC chemokines-mediated signaling [118]. In experimental in vivo systems, bortezomib inhibits neovascularization in both chick embryo chorioallantoic membrane in a dose-dependent fashion and PC-tumors in human plasmacytoma xenograft mouse [119,120,121]. In bortezomib-treated MM patients, the reduction in BM microvessel density and the reduction in circulating angiogenic cytokines levels have been demonstrated [122,123,124].

The primary target of PIs is the ubiquitin-proteasome system involved in the regulation of cellular homeostasis, cell death, and angiogenesis, both carfilzomib, a second-generation PI and ixazomib, the first orally available PI, have shown the same antiangiogenic activity [125,126].

#### 4.1.2. Immunomodulators (IMIDs)

Thalidomide, the first immunomodulatory drug, has been demonstrated to have antiangiogenic properties that increase its efficacy as an anticancer drug. Various metabolites of thalidomide have antiangiogenic activities [127,128,129]. They inhibit tubulin cytoskeletal rearrangement and filopodial formation, thus causing a reduction in the proliferation, migration, and tube formation of stimulated ECs [130]. Thalidomide and its derivatives, cause microtubules depolymerization by binding the same site of vinblastine on tubulin. Moreover, they inhibit the reassembly of microtubules, alter the dynamics of individual microtubules, decrease the growth rates, and shorten the excursions. These are the bases of the antimitotic effect of thalidomide and its derivatives because the reduction in the dynamicity of microtubules blocks cell mitosis. [131,132]. In chick and rabbit embryo limbs as well as in zebrafish embryos, thalidomide causes loss of FGF and Shh signaling [133,134,135]. Thalidomide’s antiangiogenic action is based on the destruction of vessels without smooth muscle, which typically creates a newly formed cancer vascular tree [133]. Moreover, thalidomide-induced vessel loss is due to nitric oxide expression inhibition [136,137,138], VEGF receptor depletion [139], and destruction of FGF-induced blood vessels in rodent and rabbit corneal assays [140,141]. Finally, thalidomide modulates the expression and activity of actin and tubulin, integrins, vascular endothelial growth factor, PDGFβ, nitric oxide, ceramide, angiopoietins, Notch, HIF, Slit2/Robo signaling and ROS, all molecules that are fundamental in neovessel development [142,143,144,145,146,147].

The second-generation IMiD, lenalidomide, inhibits the VEGF-induced PI3K-Akt signaling pathway (Figure 3) and HIF-1 expression [148], induces apoptosis of tumor cells, blocks the activity of TNF, and modulates T cells and NK cells activities [149,150,151,152,153]. Moreover, it inhibits MM PCs/stromal cell interaction by blocking cell adhesion [149,150,151,152,153]. It has been demonstrated that lenalidomide inhibits the proliferation and migration of ECs in patients with active MM by downregulating angiogenesis-related key genes and proteins [154]. Finally, lenalidomide inhibits in vitro neovessel formation in the Matrigel assay and in vivo PCs-induced angiogenesis in the chorioallantoic membrane (CAM) assay [154]. The anti-angiogenic effect of lenalidomide has been suggested in treated MM patients, but no significant reduction in BM neovascularization has been demonstrated [155]. 

Pomalidomide, like thalidomide and lenalidomide, inhibits experimental angiogenesis in MM ECs by targeting VEGF and HIF-1 [156]. 

#### 4.1.3. Bisphosphonates

Bisphosphonates, are antiresorptive drugs used in bone disease treatment, that have antiangiogenic activity [157]. Zoledronic acid in therapeutic doses, inhibits ECs proliferation, chemotaxis, tube forming in vitro and MM plasma cell and EC-induced angiogenesis in the in vivo CAM assay [157,158,159]. Synergistic activities in BM macrophages from MM patients have been demonstrated for bortezomib and zoledronic acid [158]. The association of these two drugs inhibits the in vitro macrophage expression of FVIII-RA, Tie2/Tek, and VEGFR-2/VE-cadherin, proliferation, adhesion, migration, and tube formation in the Matrigel assay. Moreover, bortezomib and zoledronic acid synergistically inhibit the production and release of angiogenic cytokines, VEGFR-2 expression, and phospho-activation, ERK1/2 phosphorylation, and NF-κB activation induced by VEGF-2/FGF-2 stimulation [158]. The inhibition of these biological activities blocks macrophage trans-differentiation into endothelial-like cells and inhibits the vasculogenic mimicry process [158,159].

### 4.2. Preclinical Evidence for Novel Anti-Angiogenic Inhibitors

#### 4.2.1. Monoclonal Antibodies

Evidence of the role of VEGF and its receptors in cancer progression has resulted in the development of monoclonal antibodies targeting the VEGF/VEGFR complex. In preclinical studies, the inhibition of VEGF-mediated proangiogenic activity in BM ECs derived from patients with MM has been described [72,73]. In vivo, neutralizing anti-VEGF-A antibody (Figure 3), and more efficaciously, anti-VEGFR-2 antibody (Figure 3) inhibits the constitutive autophosphorylation of both VEGFR-2 and the associated extracellular signal-regulated kinase-2 (ERK-2), proliferation, and capillarogenesis ability in MM ECs [72,73].

#### 4.2.2. Other Molecules

PTK787/ZK 222584 (PTK787), a molecule specifically designed to block the tyrosine kinase domain of VEGFR, inhibits constitutive and VEGF-induced receptor phosphorylation (Figure 3). In vitro, at clinically achievable concentrations, PTK787 can inhibit VEGF-induced cell proliferation, migration, growth, and survival, and overcome drug resistance of MM cells (cell lines and patient-derived) in the presence of Dex and IL-6 and/or cultured with BMSCs [160].

Aplidin (originally isolated from the Mediterranean tunicate Aplidium Albicans) and its analogs inhibit cell proliferation, angiogenic sprouting, and neovessel formation in vitro of human ECs [161]. Moreover, they block VEGF-induced neovascularization in the in vivo CAM system [161].

#### 4.2.3. Dual Inhibition of VEGF/cMET

There is some evidence that VEGF/VEGFR and HGF/cMet signaling are both dysregulated in MM BM ECs and have a synergistic effect in the progression of the disease [22,162,163]. Following VEGF and HGF binding to their respective receptors, dimerization and autophosphorylation of VEGFR and cMet induce the recruitment of signaling proteins to the docking site. This leads to the activation of downstream pathways such as PI3K/AKT and Ras/ERK and translates into biological responses such as cell survival, growth, migration, proliferation, metabolism, and, finally, angiogenesis [22,77,164]. These observations support the evidence that blocking a single growth factor cannot suppress the entire process of angiogenesis and may be ineffective [165]. The dual inhibition of VEGF/VEGFR and HGF/c-Met signaling may produce more satisfactory results (Figure 3). 

Pazopanib (GW786034) is a second-generation multi-targeted tyrosine kinase inhibitor against vascular endothelial growth factor receptor-1, -2, and -3, platelet-derived growth factor receptor-alpha, platelet-derived growth factor receptor-beta, and c-kit (Figure 3). In vitro pazopanib modulates the expression of surface adhesion molecule (intercellular adhesion molecule-1 and vascular cell adhesion molecule-1). This inhibits MM plasma cell and BM SCs adhesion which causes a reduction in plasma cell proliferation and migration and an increase of apoptosis. Moreover, pazopanib inhibits neovessel formation in the Matrigel assay, increases HUVEC apoptosis, and sensitizes both MM PCs to melphalan and bortezomib [166]. 

Sorafenib, a multi-kinase inhibitor that acts predominantly through inhibition of Raf-kinase and VEGF receptor 2 (Figure 3), affects the BM microenvironment and its interaction with myeloma cells by inhibition of VEGF-induced tubule formation and downregulation of VEGF and IL-6 secretion [162,163,167].

Derivatives of quinolone and quinazoline (i.e., gefitinib, erlotinib, foretinib, golvatinib, and cabozantinib) have long attracted attention because of their ability to inhibit a variety of tyrosine kinase, including c-Kit, c-Met, VEGFRs, epidermal growth factor receptor (EGFR), PDGFR, FGFR and so on [50,51] (Figure 3). So, quinolones and quinazolines are a series of promising anti-tumor compounds, especially in targeting dual c-Met and VEGFR tyrosine kinase. These small molecules are potent inhibitors of EC tubule formation, c-Met and VEGFR-2 phosphorylation, cellular invasion, and migration-disrupted tumor vasculature and promote tumor and EC death [168]. 

A new class of drugs includes a multi-domain designed ankyrin repeat protein (DARPin^®^), which simultaneously binds more target molecules. MP0250, a DARPin that simultaneously binds VEGF and HGF (Figure 3) has been investigated for its antiangiogenic activity in MM [169]. MPO250 reduces the ligand-mediated VEGFR-2 and cMET phosphorylation and inhibits the downstream signaling cascades of both receptors. In MM EC, MPO250 modulates the cytokine secretion profile and inhibits all their angiogenic activities [169]. Finally, MPO250, in combination with bortezomib, decreases the tumor burden and microvessel density in the syngeneic 5T33MM tumor model [169].

### 4.3. Clinical Experiences

#### 4.3.1. Proteasome Inhibitors

Bortezomib is active in relapsed/refractory MM patients, and in those with poor-risk cytogenetics because of its dual activity on PCs and the BM stromal compartment [170]. In this MM setting, an increase in progression-free and overall survival has been obtained when compared with dexamethasone treatment alone. No significant adverse event has been found except for reversible peripheral neuropathy and thrombocytopenia. A synergistic effect, in combination with other anti-MM drugs, is the primary goal of bortezomib treatment [170].

#### 4.3.2. Immunomodulators (IMIDs)

The major strength of thalidomide in MM patients treatment, and also for bortezomib, is dual activity on MM PCs (direct anti-MM activity) and the BM stromal compartment (immunomodulation, anti-angiogenic activity, and inhibition of the secretion of angiogenic cytokines) [146,171]. In the first trial (single-agent compassionate-use) which included 84 patients, 32% responded [172]. This led to several clinical trials in both relapsed/refractory and newly diagnosed patients [172,173,174,175,176,177], and thalidomide became the standard of care in MM patients. Also, the use of thalidomide as maintenance therapy following autologous stem cell transplantation (ASCT) has been investigated [178], and the results demonstrate a progression-free survival (PFS) benefit but no consistent overall survival (OS) benefit. Moreover, the prolonged use of thalidomide was generally limited due to toxicities [178].

Multiple different agents have been combined with thalidomide (i.e., melphalan, cyclophosphamide, liposomal doxorubicin, bortezomib, carfilzomib, elotuzumab, intensive chemotherapy regimens such as D-PACE-dexamethasone, cisplatin, doxorubicin, cyclophosphamide, etoposide) with good results in term of PFS and OS, particularly with thalidomide-containing triplets [179,180,181,182,183,184,185,186,187,188,189,190,191,192,193,194,195,196,197,198,199]. 

Lenalidomide, the second-generation IMiD, has demonstrated few of the typical thalidomide side effects [200]. Its association with dexamethasone achieves overall response rates (ORR) of 48–61% in patients previously treated with thalidomide [201,202,203]. In two studies of newly diagnosed patients, lenalidomide with dexamethasone, had an ORR of 68–91% [204,205]. Moreover, the reduced dose (“low-dose”) of dexamethasone, in combination with lenalidomide, revealed an improved side effect profile with a better one-year overall survival rate [205]. Finally, in newly diagnosed patients, a significant PFS benefit with lenalidomide in combination with bortezomib and dexamethasone has been obtained [206]. More recently, the use of lenalidomide as maintenance therapy following ASCT has also been investigated, and showed a significant OS benefit [207,208,209]. 

Multiple different agents including cyclophosphamide [210,211,212], bendamustine [213,214], melphalan [215], liposomal doxorubicin [216,217], proteasome inhibitors (bortezomib, carfilzomib, and ixazomib) [218,219,220,221,222,223,224,225], histone deacetylases (HDAC) inhibitors (panobinostat and ricolinostat) [226,227], and monoclonal antibodies (elotuzumab, daratumumab, and pembrolizumab) [228,229,230,231,232] have been combined with lenalidomide with convincing results. Other trials evaluating the association of lenalidomide with new molecules are ongoing. 

The third member of the IMID class, pomalidomide, in combination with dexamethasone, has reported a 35–63% ORR with an ORR of up to 40% in lenalidomide-refractory patients and a 37% ORR in thalidomide-refractory patients [232,233,234,235,236]. 

As for other IMIDs, the association of pomalidomide with other classes of anti-MM drugs has been shown promising results [237,238,239,240,241,242,243,244].

#### 4.3.3. Anti-VEGF Monoclonal Antibodies

Bevacizumab is a humanized monoclonal anti-VEGF antibody with a high affinity for all human VEGF isoforms [245]. Two Phase II randomized trial with bevacizumab alone or in combination with anti-MM drugs have been conducted in the relapsed/refractory MM setting. In the first trial [246], which aimed to test the efficacy and safety of bevacizumab alone and in combination with thalidomide, only 12 patients, six per arm, were enrolled. Bevacizumab-associated adverse events were reported in three patients including grade 3 hypertension, fatigue, and hyponatremia. Regarding the hematological toxicity, only one grade 3 neutropenia was reported. The addition of thalidomide did not result in more toxicity with a grade 4 pulmonary hypertension in one patient, one grade 3 fatigue, and one grade 3 lymphopenia.

The AMBER trial [247] was designed to evaluate the clinical benefit and tolerability of bevacizumab with bortezomib versus bortezomib alone and 102 patients were enrolled (49 and 53 patients, respectively). Toxicity was approximately equal in both treatment arms. Diarrhea and dehydration were more frequent in the bortezomib alone group, whereas in the bortezomib/bevacizumab group, more anemia, neutropenia, fatigue, upper respiratory tract infection, neuralgia, and hypertension were reported. No differences in peripheral neuropathy were observed.

In both these studies, the addition of bevacizumab to anti-MM therapies did not result in a significant improvement in the outcome of patients. Two partial responses (PR) and three stable diseases (SD) have been achieved in the California Cancer Consortium trial [246].

No significant differences in ORR were observed in the two treatment arms of the AMBER study [247], with 51% in the bortezomib/bevacizumab arm and 43.4% in the bortezomib alone arm. One patient per arm achieved a complete response (CR) with no significant difference in PR percentage across the two groups. The only very good partial response (VGPR) was seen in the bortezomib/bevacizumab arm (16.3% vs. 7.5% of patients). The median duration of response (DOR) was 6.9 months in the bortezomib/bevacizumab and 6.0 months in the bortezomib alone arm.

These disappointing results are related to the role played by hypoxia and other active pro-angiogenic pathways in the BM microenvironment.

#### 4.3.4. Dual Inhibition of VEGF/cMET

In a phase II trial, sorafenib taken orally at a dose of 2 × 200 mg twice daily was administered to eleven relapsed/refractory MM patients up to the completion of 13 cycles or progression [248]. One patient achieved a PR, one patient an SD, seven patients experienced disease progression (DP), one patient died early because of sepsis, and one was lost to follow-up. The median PFS was 2.6 months. The specific drug-related adverse events were mild or moderate, with only one grade 4 toxicity (cardiac infarction) [248]. Twenty-three patients were enrolled in a more extensive phase II study of sorafenib for relapsed/refractory MM patients [249]. Patients received 400 mg sorafenib twice daily for 28-day treatment cycles until progression of the disease or unacceptable toxicity. No responses were observed in this study. Only two patients with MM have been enrolled in phase I/II trial of sorafenib combined with everolimus [250] No response was observed in the two enrolled MM patients.

Vandetanib (ZD6474) is a vascular endothelial growth factor receptor (VEGFR)-2, epidermal growth factor receptor (EGFR), and rearranged during transfection (RET) tyrosine kinases inhibitor [251,252]. In a phase II study [246], vandetanib was well tolerated with only Grade 1–2 adverse events, except for rare cases of Grade 3 anemia. Despite its in vitro activity, and reduction of plasma levels of VEGF in treated MM patients, no responses or clinical benefits have been achieved.

MPO250, a bispecific VEGF/HGF-targeting DARPin^®^, was administered with bortezomib and dexamethasone to 20 highly pretreated relapsed/refractory MM patients in the MiRRoR Study [253]. All 20 patients had prior exposure to IMiDs, and PIs and nine patients received PI-based regimens as their last treatment. Preliminary results showed an ORR of 45% with one CR, three VGPR, and five PR. The median duration of response was five months (range 2–24 months).

## 5. Conclusions

In MM patients, the expression levels of VEGF in BM plasma and peripheral blood are strictly related to BM neovessel density and plasma cell infiltration. Moreover, these two parameters are correlated with the disease stage and the clinical outcome of patients. The importance of angiogenesis in MM is unquestionable, as well as the central role of VEGF in the survival, proliferation, and diffusion of PCs with paracrine and autocrine mechanisms. The interference of VEGF signaling represents a useful antiangiogenic approach in the treatment of MM. The employment of monoclonal antibody against VEGF/VEGFR and small molecule tyrosine kinase inhibitors plays a pivotal role in antiangiogenic therapy.

Although impressive preclinical results in vitro and in vivo have been obtained, the inhibition of a single proangiogenic cytokine, and in particular the use of anti-VEGF drugs alone have not performed nearly as well in MM patients. Thus, the disparity between preclinical results and the clinical ones are a significant obstacle to the development of effective anti-VEGF therapy for MM patients. These disappointing results are probably due to the vicarious action of other cytokines and signaling pathways that are active in the BM microenvironment and can bypass the block of circulating VEGF or the path started by this cytokines. The constitutive activation of VEGFR may also contribute to making the anti-VEGF antibody ineffective in humans.

Although possible modalities of resistance to blocking the VEGF/VEGFR pathway have been shown, the specific, direct molecular consequences of VEGF depletion on MM BM stromal cells, MM cell lines and primary MM PCs have also been demonstrated. This indicates that VEGF inhibition may be possible in MM patients by combiningantiangiogenic activity with the inhibition of VEGF/VEGFR signaling in the other BM stromal cells and on myeloma PCs.

Although the clinical benefits of anti-VEGF monoclonal antibodies and other anti-VEGF target molecules in MM alone do not seem to be as good as was hoped, all the reported pieces of evidence justify further research into the potential of VEGF/VEGFR inhibition for the treatment of MM patients. Future studies should address strategies for multi-target inhibitors (e.g., dual inhibitors, bispecific antibodies) as well as the combination of anti-VEGF/VEGFR inhibitors with currently used anti-MM drugs such as PIs, IMIDs, and monoclonal antibodies. Greater efficacy is emerging with drugs that simultaneously block multiple cytokines. The improved outcome of MM patients treated with the new biological drugs is also related to the critical activity in tumor microenvironment, including antiangiogenesis activity, which has an anticancer effect. So, the inhibition of cytokines pathways, which mediate the interaction between cancer cells and their microenvironment near the anti-MM activity, represents one of the primary goals of modern therapeutic approaches, and better results with regard to ORR, PFS, and OS, have been achieved in the last few years. Moreover, the evidence from preliminary results on the efficacy of the aforementioned new strategies in MM relapsed/refractory MM patients support the assumption that VEGF/VEGFR inhibition represents a useful strategy in the treatment of MM patients.

Finally, we need further studies to better elucidate the pathways involved in the BM neovascularization biological process, and the mechanisms underlying the development of microenvironment-mediated PCs resistance. Innovative approaches to bypass the vicariate pathways active after the inhibition of a single molecule, such as the use of multi-target agents or combined treatment strategies are required. The combination of multiple targeted agents could achieve the goal of maximizing the efficacy of biological treatment for MM patients. Finally, we need validated biomarkers that are different from the traditional ones (e.g., β2 microglobulin, albumin, serum, and urine protein electrophoresis, etc.) to establish personalized treatments and select responsive patient sub-populations. These markers will be identified and validated thanks to continued research in the field of microenvironment activity in cancer.

## Figures and Tables

**Figure 1 jcm-09-01765-f001:**
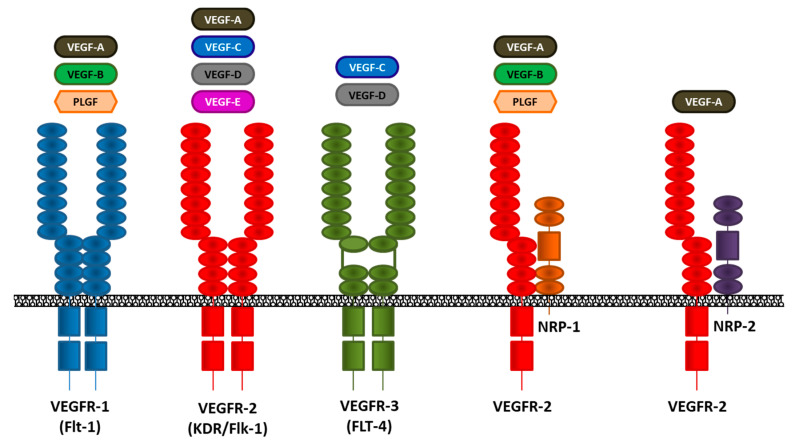
Schematic representation of VEGFRs and relative ligands. VEGFR-1 is expressed in a subset of hematopoietic bone marrow progenitor cells, monocytes/macrophages, smooth muscle cells, and several cancer cells. VEGFR-2 is expressed prevalently in vascular endothelial cells and progenitors, neurons, and megakaryocytes, but it can be expressed on the surface of several cancer cells. VEGFR-3 is restricted to lymphatic endothelial cells. Neuropilins are expressed prevalently by cells of the nervous system, and they work as co-receptors in the endothelial cell during the angiogenic process. VEGF: vascular endothelial growth factor; VEGFR: VEGF receptor; PLGF: placental growth factor; Flt-1: Fms-like tyrosine kinase-1; KDR: Kinase insert domain receptor; Flk-1: Fetal Liver Kinase-1; FLT-4: Fms-like tyrosine kinase-4; NRP: neuropilin.

**Figure 2 jcm-09-01765-f002:**
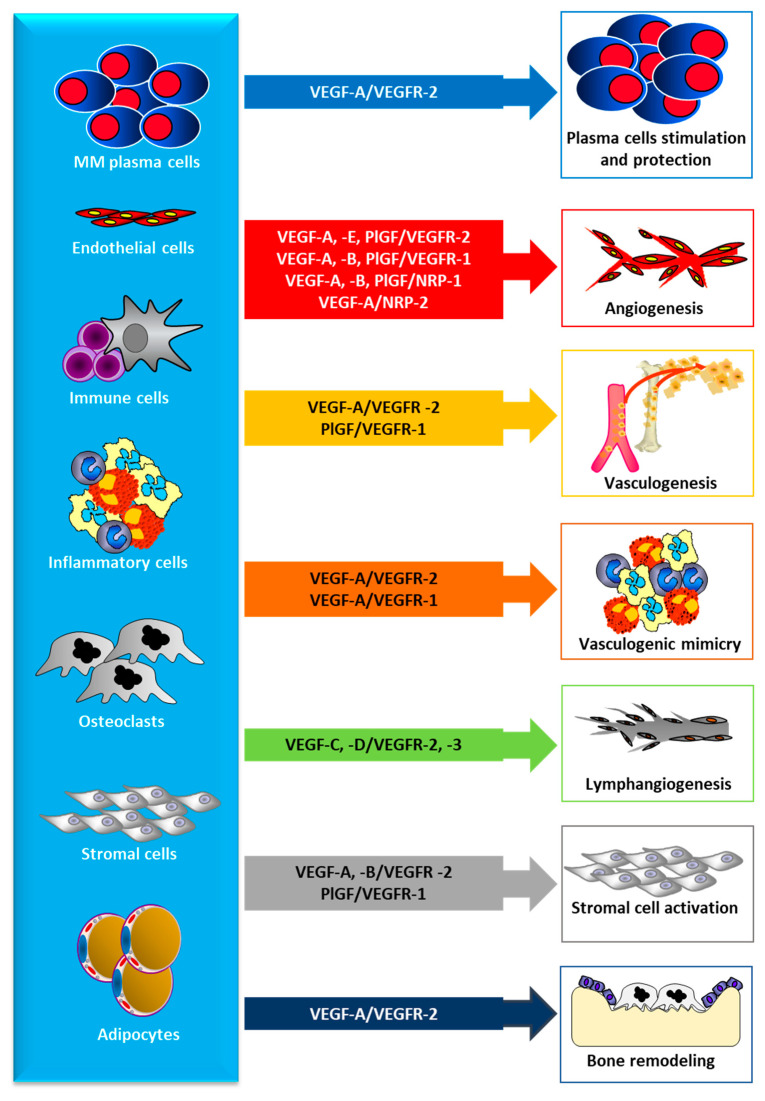
VEGFs and PlGF are is produced and released by all cells in the bone marrow microenvironment and displays several biological activities correlated to multiple myeloma (MM) progression. VEGF: vascular endothelial growth factor; VEGFR: VEGF receptor; PlGF: placental growth factor; NRP: neuropilin.

**Figure 3 jcm-09-01765-f003:**
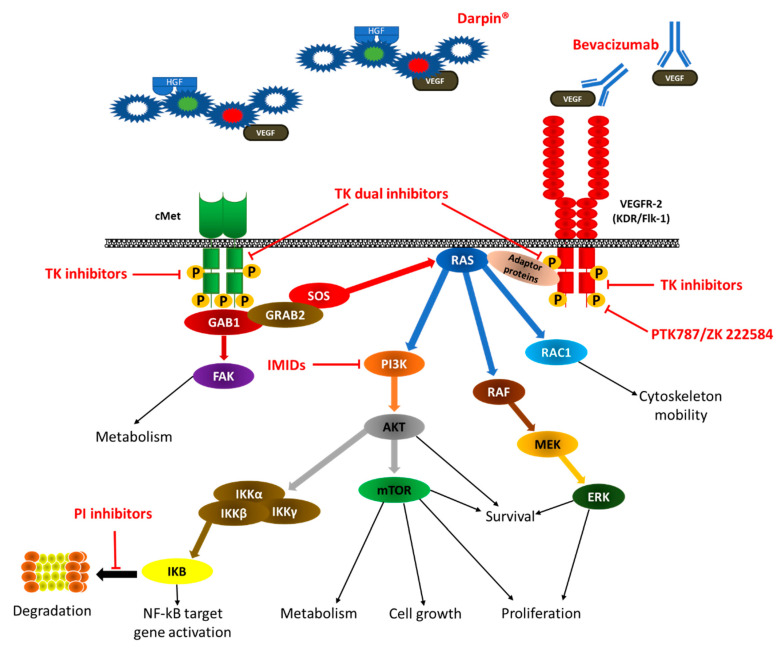
VEGFR and cMet pathways in MM microenvironment. Sites of inhibition of anti-VEGF drugs. VEGF: vascular endothelial growth factor; VEGFR: VEGF receptor; KDR: kinase insert domain receptor; Flk-1: fetal liver kinase-1; HGF: hepatocyte growth factor; cMet: tyrosine-protein kinase Met or HGF receptor; GAB1: GRB2-associated-binding protein 1; FAK: focal adhesion kinase; PI3K: phosphatidylinositol 3-kinase; AKT: also know as protein kinase B (PKB); IKK: inhibitor of nuclear factor kappa-B kinase subunits; IKB: inhibitor of nuclear factor kappa-B kinase complex; mTOR: mammalian target of rapamycin; MEK: Mitogen-activated protein kinase kinase; ERK: extracellular signal-regulated kinase; RAC1: Ras-related C3 botulinum toxin substrate 1; NF-kB: nuclear factor kappa-light-chain-enhancer of activated B cells; PI: proteasome inhibitors; IMIDs: immunomodulators; TK: tyrosine kinase.

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
