# Peer review of "Anti-VEGF Drugs in the Treatment of Multiple Myeloma Patients"

_jcm, 2020, doi:10.3390/jcm9061765_

Round 1

Reviewer 1 Report

This is a comprehensive review of VEGF inhibition in myeloma. The authors have conducted several original studies focusing on this topic for many years and they have longstanding expertise in this area. 

Minor recommendations:

  1. There have been very recent publications on the role of the bone marrow microenvironment across hematologic malignancies. Consider including for example: 

    “Bone Marrow Niches in Haematological Malignancies“

    Simón Méndez-Ferrer et al. Nat Rev Cancer.2020 May. This comprehensive review was published only days ago. Consider adding this reference and discuss findings in the current manuscript.

  2. Similarly, a recent paper using single cell sequencing of the bone marrow microenvironment in early myeloma pathogenesis (smoldering myeloma) was published ~1 week ago by the DFCI/Broad Institute. It provides new perspectives on the bone marrow microenvironment. Consider adding this reference as well and discuss findings in the current review. Here is the reference: Zavidij, O., Haradhvala, N.J., Mouhieddine, T.H. et al. Single-cell RNA sequencing reveals compromised immune microenvironment in precursor stages of multiple myeloma. Nat Cancer (2020). https://doi.org/10.1038/s43018-020-0053-3

Author Response

The Authors thank the Reviewer for helpful comments and critical suggestions.

As suggested by the Reviewer, both papers have been discussed in the revised text.

Reviewer 2 Report

Abstract:

Poor grammar.

Introduction:

  • Line 26: The MM monoclonal protein is NOT a cytokine. It is an immunoglobulin.
  • Line 28: The MM monoclonal protein is NOT responsible for the bone lesions in MM. These lesions are due to an interplay of several factors, including DKK1 and Rank-Ligand expression.
  • Line 30: the anemia of MM has not been demonstrated to be caused by alteration of osteoblasts – it is due to inflammatory changes (IL-6), renal insufficiency (less EPO), and plasma cells taking up space in the bone marrow.
  • Line 47: what does “developing of a new vascular three” mean?
  • Lines 54-55: angiogenesis has been targeted in MM in a number of negative studies. While it may be a useful target, you need to amend your concluding sentence to acknowledge that no anti-angiogenesis therapy has been shown to be effective in myeloma patients to date, including randomized trials that were conducted > 10 years ago exploring this as a treatment option.

Body of Text:

  • The authors point to bortezomib, IMiDs, etc. as useful drugs in MM due to anti-angiogenesis activity. This is very misleading as there are already other well-demonstrated anti-MM effects that these agents produce.
  • The inclusion of much preclinical data and following that with only brief reports on failed clinical trials is telling. There are many preclinical efficacy papers in MM that only go on to show no utility in the clinical setting.  Given that the VEGF question has been effectively answered for MM for many years, what is the utility of this review?

Author Response

The Authors thank the Reviewer for helpful comments and constructive criticism.

1) Abstract: Poor grammar.

The Abstract has been revised according to reviewer criticism.

2) Line 26: The MM monoclonal protein is NOT a cytokine. It is an immunoglobulin.

The authors agree with the reviewer, an error was made while correcting the first version of the text. The phrase: “The clonal plasma cells produce and release monoclonal protein cytokines…” has been corrected as: “The clonal plasma cells produce and release cytokines responsible for the typical clinical manifestation of the disease” in the revised text (lines 29, 30)

3) Line 28: The MM monoclonal protein is NOT responsible for the bone lesions in MM. These lesions are due to an interplay of several factors, including DKK1 and Rank-Ligand expression.

The authors agree with the reviewer, the correction made to the previous sentence (see criticism n. 2 - lines 29, 30) clarifies the role of cytokines, not of the monoclonal component, in the genesis of bone disease.

4) Line 30: the anemia of MM has not been demonstrated to be caused by alteration of osteoblasts – it is due to inflammatory changes (IL-6), renal insufficiency (less EPO), and plasma cells taking up space in the bone marrow.

The authors agree with the reviewer, in fact, the lines 31 and 32 of the text, at the second point, reports: “ii) anemia, caused alteration of maturation and differentiation of erythroblasts”. The authors preferred not to report extensively the causes underlying the alteration of hematopoiesis in order not to exceed in unnecessary information as not the subject of this review.

5) Line 47: what does “developing of a new vascular three” mean?

The terms “vascular three” have been corrected as “vascular tree” throughout the revised text.

6) Lines 54-55: angiogenesis has been targeted in MM in a number of negative studies. While it may be a useful target, you need to amend your concluding sentence to acknowledge that no anti-angiogenesis therapy has been shown to be effective in myeloma patients to date, including randomized trials that were conducted > 10 years ago exploring this as a treatment option.

Inhibition of a single proangiogenic cytokine (e.g. VEGF) has not been shown to be effective in the treatment of multiple myeloma, probably due to the vicarious action of other cytokines and signaling pathways. Moreover, constitutive activation of VEGFR may contribute to render ineffective the anti-VEGF antibody in human. However, greater efficacy is emerging with drugs that block multiple cytokines simultaneously. Furthermore, as shown in previous our own and other groups’ studies, the response to therapy is also related to the activity of anticancer drugs on the tumor microenvironment. Moreover, their antiangiogenic action is always well defined. So, the inhibition of cytokines production which mediates the interaction between cancer cells and their microenvironment represents one of the major goals of the modern therapeutic approaches. Based on these considerations, according to the reviewer criticism, the last paragraph of the introduction has been modified as:  “Taken together, these observations led to consider the modifications in the BM microenvironment and its controllers (activated cells, cytokines and their autocrine and paracrine loops, signaling pathways) useful targets in the treatment of MM near the direct targeting of MM plasma cells.” in the revised text (lines 61-64).

7) Body of Text: The authors point to bortezomib, IMiDs, etc. as useful drugs in MM due to anti-angiogenesis activity. This is very misleading as there are already other well-demonstrated anti-MM effects that these agents produce.

As reported in the previous response (point 6), the major goal of the new and newest drugs is the biological targeting of MM plasma cells in their microenvironment. This is a well-demonstrated activity of proteasome inhibitors, IMIDs, and other new non-chemotherapeutic agents. Moreover, the antiangiogenic activity of these drugs is well demonstrated in experimental conditions (in vitro and in vivo) as well as in treated patients. Finally, their direct or indirect inhibition of the VEGFs/VEGFRs pathways is also demonstrated and reported in several works of our group and other authors. This point has been extensively reported in the conclusion of the revised text.

8) The inclusion of much preclinical data and following that with only brief reports on failed clinical trials is telling. There are many preclinical efficacy papers in MM that only go on to show no utility in the clinical setting.  Given that the VEGF question has been effectively answered for MM for many years, what is the utility of this review?

As reported in previous responses, as shown in previous our own and other groups’ studies, the response to therapy is related to the activity of anticancer drugs also on the tumor microenvironment. In particular, the antiangiogenic action is always well defined. So, the inhibition of cytokines production and activity, including VEGF, which mediates the interaction between cancer cells and their microenvironment represents one of the major goals of the modern therapeutic approaches. The simultaneous inhibition of more than one pathway, for example VEGF/VEGFR and HGF/cMet pathways, has demonstrated good activity in recent trials. This point has been discussed throughout the text.

Reviewer 3 Report

  1. Tell more (3-4 sentences) about mast cell  phenotypic adaptation to vasculogenesis mimicry (line 95 and 158).
  2. Is  vasculogenic mimicry   dependent from inflammatory cytokines( i.e. IL-6, TGFalpha,  IL-8) expression ? ( line 154-161)
  3. It seems to be indicated to tell more (3-5 sentences) about mechanisms of actine and tubulin expression influenced by thalidomide administration (line 273-274) 
  4. Please show the name of " proteasome inhibitors "and "monoclonal antibodies" combined with lenalidomide (line 390 and 391)
  5. In "Conclusion" it seems to be necessary to tell more about "further studies on angiogenesis , pro-angiogenic pathways  and personalized treatment.( 5 -6 sentences) 

Author Response

The Authors thank the Reviewer for helpful comments and useful suggestions.

1) Tell more (3-4 sentences) about mast cell phenotypic adaptation to vasculogenesis mimicry (line 95 and 158).

According to the reviewer's suggestion,  the phrases: “They maintain typical tryptase-positivity and acquire endothelial cell-like properties. In the BM of MM patients, mast cells interact physically with ECs participating in the completion of the vascular wall” and the relative reference have been added in the revised text (lines 109-111).

2) Is vasculogenic mimicry   dependent from inflammatory cytokines (i.e. IL-6, TGF-alpha, IL-8) expression? (line 154-161)

The vasculogenic mimicry is a pathological process in which epithelial-mesenchymal transition plays a very important role. Cancer cells, cancer stem cells, inflammatory cells, tumor-associated fibroblasts, under the stimulus of angiogenic cytokines (FGF-2, VEGF-A, HGF, IGF.), inflammatory cytokines (IL-6 and -8, TGF-α and -β, TNF-α, CCL-2), and other signaling pathways (SDF1, COX-2, Twist, E-cadherin, EphA2-PI3K, Wnt, and Notch signaling pathway) undergo to epithelial-mesenchymal transition by mean downregulation of E-cadherin, zonula occludins-1, and α-catenin, and upregulation of VE-cadherin, fibronectin, cadherin-2, and vimentin. Also hypoxia-inducible factors (HIFs) and hypoxia-responsive elements (HREs) play an important role in this context. These phrases have been inserted in the revised text (lines 172-180).

3) It seems to be indicated to tell more (3-5 sentences) about mechanisms of actine and tubulin expression influenced by thalidomide administration (line 273-274)

According to the reviewer's criticism, the sentences: “Thalidomide and its derivatives, by binding the same site of vinblastine on tubulin, cause microtubules depolymerization. Moreover, they inhibit the microtubules reassemblation alterating the dynamics of individual microtubules, decreasing the growth rates, and shortening the excursions. These are is the bases of the antimitotic effect of thalidomide and its derivatives because of the reduction in the dynamicity of microtubules that block cell mitosis.” have been addressed in the revised text (lines 304-309).

4) Please show the name of " proteasome inhibitors "and "monoclonal antibodies" combined with lenalidomide (line 390 and 391)

As suggested by the Reviewer, we have reported the names of all drugs combined with lenalidomide in the revised text (lines 432-436).

5) In "Conclusion" it seems to be necessary to tell more about "further studies on angiogenesis, pro-angiogenic pathways and personalized treatment. (5 -6 sentences)  

As suggested by the reviewer the paragraph “conclusion has been revised in the new text.

Reviewer 4 Report

The review about VEGF Inhibitors in the Treatment of MM patients is very complex and comprehensive. Although it contains a lot of information, it is a bit dry, with multiple grammar and spelling errors. The authors should in particular correct mistakes in the words: vascular tree (not vascular three). Also, please check for correct use and explanation of abbreviations, such as MGUS (explained in a line 63 and 124).

Moreover, on the page 4, line 166, there I believe should be in brackets indicated interactions between VEGF and VEGFR (such as VEGF-C, -D/VEGFR-2, -3), not (such as VEGFR-C, -D/VEGFR-2, -3),

Figure 1: Please to the figure legend add description of the abbreviations used in the figure. In the figure it would be beneficial to describe the cell types on which the particular receptor/ligand interaction are present.

The amount of information about different VEGF signaling should be present also in a more digestible way in a figure. Thus, in figure 2, describe also the signaling pathways that VEGF activates in respective processes described here (such as angiogenesis, vasculogenesis, etc…).

The authors describe preclinical evidence of anti-angiogenic effects of currently used drugs in MM and novel therapies used for VEGF inhibition. This part is a bit chaotic, as the anti-angiogenic effect of e.g. bortezomib is a side-effect or a downstream effect of proteasome inhibition, not a clear on-target effect, as for example in the case of monoclonal antibodies against VEGF/VEGFR complex. Thus, I suggest dividing the part into e.g.: anti-angiogenic effect of currently used therapy (proteasome inhibitors, IMiDs, bisphosphonates) and novel anti-angiogenic inhibitors (moAb, other molecules, dual inhibitors). Furthermore, the authors present here 3 generations of IMiDs, but only first generation PI bortezomib, is there known anything more to carfilzomib or ixazomib?

In conclusion, can the authors speculate, why the anti-VEGF drugs are not performing as well in patients, although they are very promising in vitro? Are there any known mechanism of resistance in patients that are not present in vitro?

Author Response

The Authors thank the Reviewer for helpful comments and useful suggestions.

1) The review about VEGF Inhibitors in the Treatment of MM patients is very complex and comprehensive. Although it contains a lot of information, it is a bit dry, with multiple grammar and spelling errors. The authors should in particular correct mistakes in the words: vascular tree (not vascular three). Also, please check for correct use and explanation of abbreviations, such as MGUS (explained in a line 63 and 124).

The authors agree with the reviewer. The terms “vascular three” have been corrected as “vascular tree” throughout the revised text. Moreover, the correct use and explanation of abbreviations has also revised.

2) Moreover, on the page 4, line 166, there I believe should be in brackets indicated interactions between VEGF and VEGFR (such as VEGF-C, -D/VEGFR-2, -3), not (such as VEGFR-C, -D/VEGFR-2, -3),

According to the reviewer criticism, the terms: “(VEGFR-A, -E/VEGFR-2-neuropilin-1)”,  “(VEGFR-C, -D/VEGFR-2, -3)”, and “VEGFR-B/VEGFR-1”, have been corrected as: “(VEGFR-A, -E/VEGFR-2-neuropilin-1)”, “(VEGF-C, -D/VEGFR-2, -3)”, and “VEGF-B/VEGFR-1” respectively,  in the revised text (lines 189, 190).

3) Figure 1: Please to the figure legend add description of the abbreviations used in the figure. In the figure it would be beneficial to describe the cell types on which the particular receptor/ligand interaction are present.

As suggested by the Reviewer, the description of the abbreviations used in figure 1 and the cell types on which the receptor are mainly present have been reported in the legend to the figure. The receptor/ligand interaction involved in angiogenesis are described throughout the text.

4) The amount of information about different VEGF signaling should be present also in a more digestible way in a figure. Thus, in figure 2, describe also the signaling pathways that VEGF activates in respective processes described here (such as angiogenesis, vasculogenesis, etc…).

As suggested by the Reviewer, figure 2 has been modified in the revised text.

5) The authors describe preclinical evidence of anti-angiogenic effects of currently used drugs in MM and novel therapies used for VEGF inhibition. This part is a bit chaotic, as the anti-angiogenic effect of e.g. bortezomib is a side-effect or a downstream effect of proteasome inhibition, not a clear on-target effect, as for example in the case of monoclonal antibodies against VEGF/VEGFR complex. Thus, I suggest dividing the part into e.g.: anti-angiogenic effect of currently used therapy (proteasome inhibitors, IMiDs, bisphosphonates) and novel anti-angiogenic inhibitors (moAb, other molecules, dual inhibitors).

In the revised text the part of “preclinical pieces of evidence” has been divided according to the reviewer's suggestion.

6) Furthermore, the authors present here 3 generations of IMiDs, but only first generation PI bortezomib, is there known anything more to carfilzomib or ixazomib?

The primary target of all the PIs is the ubiquitin-proteasome system. This is a component of all eukaryotic cells and is involved in the regulation of cellular homeostasis and cell death. The antiangiogenic effect of all the PIs is due to NFk-B block through proteasome inhibition. So, both the second-generation PI carfilzomib and the oral available PI ixazomib have the same antiangiogenic effect of bortezomib. These pieces of evidence have been discussed in the revised text (lines 297-299)

7) In conclusion, can the authors speculate, why the anti-VEGF drugs are not performing as well in patients, although they are very promising in vitro? Are there any known mechanism of resistance in patients that are not present in vitro?

Inhibition of a single proangiogenic cytokine (e.g. VEGF) has not been shown to be effective in the treatment of multiple myeloma, probably due to the vicarious action of other cytokines and signaling pathways (e.g. HGF/cMet or integrins pathways). Moreover, constitutive activation of VEGFR may contribute to render ineffective the anti-VEGF antibody in human. However, greater efficacy is emerging with drugs that block multiple cytokines simultaneously. Furthermore, as shown in previous our own and other groups’ studies, the response to therapy is also related to the activity of anticancer drugs on the tumor microenvironment. Moreover, their antiangiogenic action is always well defined. So, the inhibition of cytokines production which mediates the interaction between cancer cells and their microenvironment represents one of the major goals of the modern therapeutic approaches. Based on these considerations, according with the reviewer criticism, the last paragraph of the introduction has been modified as:  “Taken together, these observations led to consider the modifications in the BM microenvironment and its controllers (activated cells, cytokines and their autocrine and paracrine loops, signaling pathways) useful targets in the treatment of MM near the direct targeting of MM plasma cells.” in the revised text (lines 61-64), and all these considerations have been reported in the conclusion paragraph.

Round 2

Reviewer 2 Report

The authors did not significantly address prior concerns regarding clinical utility of targeting vegf in MM. 

Author Response

The Authors thank the Reviewer for helpful comments and critical suggestions.

In MM patients, the expression levels of VEGF in BM plasma and peripheral blood are strictly related to BM neovessel density and plasma cell infiltration. Moreover, these two parameters correlated with the disease's stage and the clinical outcome of patients. The importance of angiogenesis in MM is unquestionable, as well as the central role of VEGF in survival, proliferation, and diffusion of plasma cells with paracrine and autocrine mechanisms. The interference with VEGF signaling represents a useful antiangiogenic approach in the treatment of MM. The employment of monoclonal antibody against VEGF/VEGFR and small molecule tyrosine kinase inhibitors plays a pivotal role in antiangiogenic therapy.

A significant obstacle to the development of effective anti-VEGF therapy for MM patients resides in the disparity between preclinical results and the clinical ones.

Moreover, the specific molecular consequences of VEGF depletion directly on MM BM stromal cells, which is similar to those obtained on MM cell lines and primary MM patients plasma cells, have been demonstrated. Thus indicates that VEGF inhibition may be useful in MM patients by mean a combination of the antiangiogenic activity with the inhibition of VEGF/VEGFR signaling in the other BM stromal cells and on myeloma plasma cells. Although the clinical benefits of anti-VEGF monoclonal antibodies and other anti-VEGF target molecules in MM alone seem to be not as active as was hoped, all the reported pieces of evidence justify further research into the potential of VEGF/VEGFR inhibition for the treatment of MM patients. Future studies should address strategies on multitarget inhibitors (e.g., dual inhibitors, bispecific antibodies) as well as on the combination of anti-VEGF/VEGFR inhibitors with currently used anti-MM drugs such as PIs, IMIDs, monoclonal antibodies.15 for instance). The shreds of evidence of the preliminary results on the efficacy of the aforementioned new strategies in MM relapsed/refractory MM patients support the assumption that VEGF/VEGFR inhibition represents a useful strategy in the treatment of MM patients.

According to the Reviewer’s criticism, all these sentences have been reported in the conclusion paragraph.

Reviewer 4 Report

The authors have improved the manuscript according to the previous suggestions.

I still suggest language correction.

Author Response

The Authors thank the Reviewer for helpful comments and critical suggestions.

As suggested by the Reviewer, the paper has been revised for language correction.